# Business Dynamism and Innovation Capacity, an Entrepreneurship Worldwide Perspective

**João Lopes** [1,2], **Márcio Oliveira** [1,3,*], **Paulo Silveira** [4], **Luís Farinha** [1,5] **and José Oliveira** [6]

1   NECE—Research Center in Business Sciences, University of Beira Interior, 6201-001 Covilha, Portugal;
    joao.lopes.1987@hotmail.com (J.L.); luis.farinha@ipcb.pt (L.F.)
2   Miguel Torga Institute of Higher Education, 3000-132 Coimbra, Portugal
3   School of Education and Social Sciences, Polytechnic Institute of Leiria, 2411-901 Leiria, Portugal
4   SHERU—Sport, Health and Exercise Research Unit, Polytechnic Institute of Castelo Branco,
    6000-266 Castelo Branco, Portugal; paulo.silveira@ipcb.pt
5   CIPEC—Research Center in Heritage, Education and Culture, Polytechnic Institute of Castelo Branco,
    6060-163 Idanha-a-Nova, Portugal
6   ALGORITMI Research Centre, University of Minho, 4704-553 Braga, Portugal; jcastroliveira@gmail.com
*   Correspondence: marcio.oliveira@ipleiria.pt

**Abstract:** This research aims to identify which factors best explain business dynamics and innovation capacity in the continents of Africa, Asia, Europe, Latin America and the Caribbean, and North America. To achieve this, data from the Global Entrepreneurship Monitor and the Global Competitiveness Report is used. The linear regression method is utilized with the stepwise procedure for data analysis. It is possible to ascertain that, with a view to increasing innovation capacity in the African continent, business leaders and managers should be acquainted with innovation studies to better understand technological advances. In relation to Asia, the detected models of business dynamism and capacity for innovation are positive. On the European continent, the results show that RIS3 has a positive impact on the capacity for innovation. In Latin America and the Caribbean, it seems that business dynamism and the capacity for innovation are negative and regional development policies should be more flexible. In North America, it appears that business dynamism and the capacity for innovation are negative. The research contributes with measures that can be applied by organizations and policymakers to these five continents to improve the performance of business dynamism and the capacity for innovation in their territories. The resulting data give originality to the research as well as important contributions, not only to the theory, but also to the entities (organizations and governments) acting in the field who can implement new policies, such as tax incentives to companies for the first purchase of high-tech equipment, products, or products with intellectual property rights developed by national companies and provide support policies directed to companies that purchase high-tech domestic equipment.

**Keywords:** dynamism; innovation capability; open innovation; GEM; GCR; policies; entrepreneurship

## 1. Introduction

The notion of entrepreneurship, debated in today's management studies, descends from Richard Cantillion's initial approach in 1755, as a consequence, the term "entrepreneur" was introduced for the first time [1]. The concept of the entrepreneur, as being a person who innovates was introduced later by Joseph Schumpeter in 1934 [2]. Nowadays, the concept of entrepreneurship is extensively spread amongst the knowledge areas of political and social sciences, such as history; psychology; anthropology; economy; sociology, or political science. Entrepreneurship is studied in different manners and encompasses interdisciplinary theories and approaches [3,4].

The Global Entrepreneurship Monitor (GEM) model aims to articulate a coherent view on national standards for generating new opportunities and creating businesses, which can impact local economies [5–7].

The purpose of this research is to fulfill the gap identified by [8]. The author suggests utilizing GEM data to discover relationships and causal models which may clarify the phenomenon of entrepreneurship in distinctive countries or regions. According to [9], when countries or regions are investigated independently, heterogeneity is discovered, exposing indirect evidence of the importance of political-institutional factors, determinants of entrepreneurship.

Therefore, this research aims to present explanatory models for the continents of Africa, Asia, Europe, Latin America and the Caribbean, and North America with regard to business dynamism and innovation capability. Data was collected from the Global Entrepreneurship Monitor (GEM) and the Global Competitiveness Report (GCR). For this purpose, the linear regression method was used with the stepwise procedure for data analysis.

From the results obtained, it is possible to see that the European and Asian continents are the only ones that do not have variables that affect negatively the business dynamism. However, Europe has more variables that positively affect business dynamism than Asia. Regarding the African continents, Latin America and the Caribbean, and North America, it seems that Latin America and the Caribbean are the ones to present a higher number of variables that positively affect business dynamism. The continent where business dynamism is most negative is North America.

Regarding innovation capability, we found that it is negative on all continents except in Asia. North America is the continent with the most negative innovation capability. On the other hand, Latin America and the Caribbean have more variables that affect positively innovation capability.

The collected data allow us to realize that there is a set of measures that can be applied by organizations and policymakers on these five continents, thus contributing to improving the performance of business dynamism and the capacity for innovation. Amongst these measures, we highlight the attribution of tax incentives to companies to purchase equipment or high-tech products, or products with intellectual property rights developed by national companies, as well as the provision of support policies directed to companies that purchase high-tech household equipment.

This research is organized into five parts. Initially, an introduction is made, followed by a literature review. In the third part, the methodology used is described. In parts four and five the results are presented and discussed. Finally, conclusions, limitations, and future lines of investigation are presented.

## 2. Literature Review in Business Dynamics and Innovation Capability in Entrepreneurship

Currently, entrepreneurship is understood as a significant factor of productivity, innovation, employment, and competitiveness. Entrepreneurship is essential to economic dynamism and strongly impacts the economic growth of countries and regions [10–12]. The establishment of new businesses and entrepreneurship are key components in the regional economic growth as they influence the regional industrial base configuration, this is an important indicator to reveal the growth differences and performance between regions [13].

Social, cultural, political, and economic contexts can positively or negatively affect entrepreneurial activity. For example, in Western European countries, such as Spain, Portugal, or Italy, where the unemployment rate is traditionally high, this data can serve as an incentive for an increase in entrepreneurial activity in these countries [14,15]. In another example, we can mention the countries of Latin America, where there has been significant progress in terms of democracy, property rights, and macroeconomic stability, data that coexist with fragile educational systems, knowledge creation, or even deficient economic reforms. In this sense, these regions have shown greater difficulties in improving economic performance in comparison with other emerging markets [16–18].

When one intends to understand the dynamics of businesses that are introduced in constantly changing environments, the static view of a business model of a company or

organization is not adequate or sufficient [19]. It is necessary to analyze the impulses and their intensity impending from the inside and outside of the organizations, which ultimately will affect their performance in different ways [20].

According to the authors of [21], business dynamics are like a complex system of interrelated sub-components of value creation, interacting with heterogeneous internal and external influences leading to the evolution of its components and the system itself. Doing business in these dynamic contexts can lead to transformative innovations. These business dynamics can even be seen as a fundamental skill that allows companies or organizations to achieve long-term sustainability [22].

Distinctive values, abilities, priorities, and attitudes distinguish the entrepreneurial society from others. In the entrepreneurial society, and just like business dynamism, entrepreneurship is the basis of economic growth, employment creation, and competitiveness [23].

Business dynamism is the procedure by which companies are born, "die", expand, and compress constantly [24].

While business dynamics refers to an attitude towards a context, innovation capacity focuses on the ability to shape and manage various capacities, with organizations managing innovative capacity being able to integrate important capacities and resources in order to promote innovation successfully [25].

Economic performance is also influenced by innovation capability. Thus, innovation capability consists of encouraging collaboration, connectivity, creativity, diversity, and confrontation between different perspectives in a given region or country. Innovation capability also includes the ability to put innovative ideas into practice that are transformed into new goods and services [19], but it can also be studied or directed to different areas of management, such as, the design of products or services, new processes, or new ways of co-communicating [26].

The capacity for innovation promotes competitiveness, growth, and performance [27], being so, they need to dynamically deploy, mobilize, integrate and align their resources and capacities to innovate and obtain competitive advantage [28].

Thus, the capacity for innovation and business dynamics becomes essential for companies to be able to face the turbulent and rapidly changing environment [29] and manage the entrepreneurship so necessary for the sustainable development of regions, countries, and continents.

The GEM describes the characteristics of entrepreneurship [30], recognizing the behaviors of individuals, entrepreneurs as proactive, innovative, with the ability to take risks responsibly and always in interaction with the environment [31]. GEM takes into account the relationship and interdependence between entrepreneurship and the development of the economy, in order to understand the factors that stimulate or hinder entrepreneurship [32,33]. GEM also provides an evaluation platform, as entrepreneurs, in their activity, influence economic growth where, as if in a vicious cycle, entrepreneurial capacity is strengthened even more [10,30].

## 3. Methodology

The purpose of this research is to present explanatory models for the continents of Africa, Asia, Europe, Latin America and the Caribbean, and North America with regards to business dynamism and innovation capability. It was not possible to include Oceania because there is no data available for the models previously indicated. Data were collected on 15 November 2018 at GEM (www.gemconsortium.org/data, accessed on 18 November 2020) and GCR 2018 (www.weforum.org/reports/the-global-competitveness-report-2018, accessed on 18 November 2020). A total of 52 countries were included in the present research, which were simultaneously available in the indicated databases (Table 1). The number of countries considered for the research was 52 since this was the number of countries that coincided in the two databases consulted.

**Table 1.** Countries included in the research.

| Africa | Asia | Europe | Latin America and the Caribbean | North America |
|---|---|---|---|---|
| Egypt | China | Bosnia and Herzegovina | Argentina | Canada |
| Morocco | India | Bulgaria | Brazil | United State |
| South Africa | Indonesia | Croatia | Chile | - |
| - | Iran | Cyprus | Colombia | - |
| - | Israel | Estonia | Ecuador | - |
| - | Japan | France | Guatemala | - |
| - | Kazakhstan | Germany | Mexico | - |
| - | Lebanon | Greece | Panama | - |
| - | Malaysia | Ireland | Peru | - |
| - | Qatar | Italy | Uruguay | - |
| - | Saudi Arabia | Latvia | - | - |
| - | South Korea | Luxembourg | - | - |
| - | Taiwan | Netherlands | - | - |
| - | Thailand | Poland | - | - |
| - | United Arab Emirates | Slovakia | - | - |
| - | Vietnam | Slovenia | - | - |
| - | - | Spain | - | - |
| - | - | Sweden | - | - |
| - | - | Switzerland | - | - |
| - | - | United Kingdom | - | - |
| 3 Countries | 16 Countries | 20 Countries | 10 Countries | 2 Countries |

Collected data were inserted into Microsoft Excel, and then imported and analyzed using SPSS 26.0 software.

*Method and Variables in Analysis*

The linear regression method was used with the stepwise procedure for data analysis. The multiple linear regression method is advisable because it allows to describe each continent, the relationship between each of the dependent variables: "Business Dynamism" and "Innovation Capability" and a set of independent variables: Perceived Opportunities, Perceived Capabilities, Cultural and Social Norms, Internal Market Dynamics, Physical and Services Infra-structure, Post School Entrepreneurial Education and Training, R&D Transfer, Internal Market Openness, Commercial and Professional Infrastructure, Financing for Entrepreneurs, Female/Male TEA, Female/Male Opportunity-Driven TEA, High Job Creation Expectation, Entrepreneurial Employee Activity, High Status To Successful Entrepreneurs, Fear of Failure Rate, Innovation, Business Services Sector, Total Early-Stage Entrepreneurial Activity (TEA), Governmental Support and Policies, Taxes and Bureaucracy Governmental Programs, Established Business Ownership, Entrepreneurship as a Good Career Choice, Motivational Index, Entrepreneurial Intentions, and Basic School Entrepreneurial Education and Training. In addition, the linear regression models allow each continent to forecast each of the dependent variables as a function of the values of the independent variables of the models.

The validity of the models is satisfactory and justified by the values of the coefficient of determination (R square) found for each model and on each continent, which also justifies the use of regression methods. This coefficient represents the proportion of the variability of the dependent variables that is explained by the regression model.

Based on these independent variables, we estimate the parameters for the models of the five continents (Africa, Asia, Europe, Latin America and the Caribbean, and North America) taking the variable "Business dynamism" as the dependent variable.

The process was repeated with the same independent variables and considering the variable "innovation capability" as a dependent variable, also for each of the five continents.

Whenever the stepwise method introduces a new variable in the model, the significance of each variable is analyzed before eliminating any variables that do not give any significant explanatory capacity. The process is repeated until the variables not included in the model have no significant explanation, whilst all the variables included in the model have it.

Therefore, in order to verify each of the models and for each continent, we have to exclude variables that do not have statistical significance (sig < 0.05) one by one, always starting with the variable that, statistically, has less importance for the model.

## 4. Results

We start by analyzing the models obtained, in the SPSS, for each continent and considering the "business dynamism" variable as a dependent variable (Table 2).

**Table 2.** Business dynamism models.

| Continent | Model | R | R Square | Adjusted R Square | Std. Error of the Estimate |
|---|---|---|---|---|---|
| Africa | 1 | 0.982 [a] | 0.963 | 0.945 | 6.66133 |
| Asia | 1 | 0.677 [c] | 0.458 | 0.417 | 6.76614 |
| | 2 | 0.818 [d] | 0.668 | 0.613 | 5.51043 |
| Europe | 1 | 0.701 [e] | 0.492 | 0.463 | 6.13830 |
| | 2 | 0.791 [f] | 0.626 | 0.582 | 5.41680 |
| | 3 | 0.859 [g] | 0.739 | 0.690 | 4.66902 |
| Latin America and the Caribbean | 1 | 0.762 [h] | 0.581 | 0.529 | 15.17346 |
| | 2 | 0.933 [i] | 0.870 | 0.833 | 9.02722 |
| | 3 | 0.978 [j] | 0.956 | 0.933 | 5.70036 |
| North America | 1 | 1.000 [k] | 1.000 | | |

[a]. Predictors: (Constant), Established Business Ownership; [c]. Predictors: (Constant), Financing for Entrepreneurs; [d]. Predictors: (Constant), Financing for Entrepreneurs, Business Services Sector; [e]. Predictors: (Constant), Business Services Sector; [f]. Predictors: (Constant), Business Services Sector, High Status to Successful Entrepreneurs; [g]. Predictors: (Constant), Business Services Sector, High Status to Successful Entrepreneurs, Female/Male TEA; [h]. Predictors: (Constant), Cultural and Social Norms; [i]. Predictors: (Constant), Cultural and Social Norms, Taxes and Bureaucracy; [j]. Predictors: (Constant), Cultural and Social Norms, Taxes and Bureaucracy, Post School Entrepreneurial Education and Training; [k]. Predictors: (Constant), Taxes and Bureaucracy.

For the final model with the dependent variable business dynamism, the influence statistics cannot be calculated because the fit is perfect for Mainland North America.

In each continent, we have to consider the R square of the superior model obtained in the stepwise method. Thus, the R Square for the best model obtained for each of the remaining five continents (Africa, Asia, Europe, Latin America and the Caribbean, and North America) is, respectively, 0.963, 0.668, 0.739, 0.956, 1.0, which represents satisfactory values.

In Table 3, we present the coefficients for each of the continents that allow writing the linear regression equation for the best model obtained.

Based on the results in Table 3, we elaborated the linear regression equations for the dependent variable business dynamism for the five continents.

The linear regression equation for Africa is:

$$\text{Business dynamism} = 69.755 - 2.299 \times \text{Established Business Ownership}$$

The linear regression equation for Asia is:

$$\text{Business dynamism} = 25.794 + 12.197 \times \text{Financing for Entrepreneurs} + 0.436 \times \text{Business Services Sector}$$

The linear regression equation for Europe is:

$$\text{Business dynamism} = 0.845 + 0.694 \times \text{Business Services Sector} + 0.520 \times \text{High Status to Successful Entrepreneurs} + 24.248 \times \text{Female/Male TEA}$$

The linear regression equation for Latin America and the Caribbean is:

$$\text{Business dynamism} = -188.913 + 31.101 \times \text{Cultural and Social Norms} + 35.064 \times \text{Taxes and Bureaucracy} + 22.388 \times \text{Post School Entrepreneurial Education and Training}$$

The linear regression equation for North America is:

$$\text{Business dynamism} = -190.845 + 89.545 \times \text{Taxes and Bureaucracy}$$

**Table 3.** Business dynamism coefficients.

| Continent | | Model | Unstandardized Coefficients | | Standardized Coefficients | t | Sig. |
|---|---|---|---|---|---|---|---|
| | | | B | Std. Error | B | | |
| Africa | 1 | (Constant) | 69.755 | 5.034 | | 13.857 | 0.005 |
| | | Established Business Ownership | −2.299 | 0.317 | −0.982 | −7.261 | 0.018 |
| Asia | 1 | (Constant) | 33.185 | 10.102 | | 3.285 | 0.006 |
| | | Financing for Entrepreneurs | 11.596 | 3.495 | 0.677 | 3.318 | 0.006 |
| | 2 | (Constant) | 25.794 | 8.653 | | 2.981 | 0.011 |
| | | Financing for Entrepreneurs | 12.197 | 2.855 | 0.712 | 4.272 | 0.001 |
| | | Business Services Sector | 0.436 | 0.158 | 0.460 | 2.757 | 0.017 |
| Europe | 1 | (Constant) | 48.253 | 4.934 | | 9.779 | 0.000 |
| | | Business Services Sector | 0.784 | 0.188 | 0.701 | 4.173 | 0.001 |
| | 2 | (Constant) | 27.137 | 9.586 | | 2.831 | 0.012 |
| | | Business Services Sector | 0.653 | 0.174 | 0.584 | 3.755 | 0.002 |
| | | High Status to Successful Entrepreneurs | 0.363 | 0.147 | 0.385 | 2.473 | 0.024 |
| | 3 | (Constant) | 0.845 | 12.989 | | 0.065 | 0.949 |
| | | Business Services Sector | 0.694 | 0.151 | 0.621 | 4.602 | 0.000 |
| | | High Status to Successful Entrepreneurs | 0.520 | 0.140 | 0.551 | 3.715 | 0.002 |
| | | Female/Male TEA | 24.248 | 9.243 | 0.381 | 2.623 | 0.018 |
| Latin America and the Caribbean | 1 | (Constant) | −80.055 | 40.776 | | −1.963 | 0.085 |
| | | Cultural and Social Norms | 43.068 | 12.933 | 0.762 | 3.330 | 0.010 |
| | 2 | (Constant) | −110.535 | 25.457 | | −4.342 | 0.003 |
| | | Cultural and Social Norms | 33.289 | 8.083 | 0.589 | 4.119 | 0.004 |
| | | Taxes and Bureaucracy | 28.014 | 7.092 | 0.565 | 3.950 | 0.006 |
| | 3 | (Constant) | −188.913 | 28.108 | | −6.721 | 0.001 |
| | | Cultural and Social Norms | 31.101 | 5.144 | 0.550 | 6.046 | 0.001 |
| | | Taxes and Bureaucracy | 35.064 | 4.935 | 0.707 | 7.105 | 0.000 |
| | | Post School Entrepreneurial Education and Training | 22.388 | 6.586 | 0.322 | 3.399 | 0.015 |
| North America | 1 | (Constant) | −190.845 | 0.000 | | | |
| | | Taxes and Bureaucracy | 89.545 | 0.000 | 1.000 | | |

The application of linear regression verified the assumptions of normality demonstrated by the graphs of the normal probability of the residuals and by the Kolmogorov–Smirnov test, where *p*-values of 0.062, 0.748, 0.627, 0.055 were obtained for each of the continents (Africa, Asia, Europe, Latin America and the Caribbean).

We now proceed to analyze the models obtained, in SPSS, for each continent and considering the "innovation capability" as a dependent variable (Table 4).

**Table 4.** Innovation capability models.

| Continent | Model | R | R Square | Adjusted R Square | Std. Error of the Estimate |
|---|---|---|---|---|---|
| Africa | 1 | 0.996 [a] | 0.992 | 0.988 | 2.15916 |
| | 2 | 1.000 [b] | 1.000 | 1.000 | 0.01354 |
| | 3 | 1.000 [c] | 1.000 | | |
| Asia | 1 | 0.718 [e] | 0.515 | 0.478 | 12.39254 |
| Europe | 1 | 0.720 [f] | 0.518 | 0.491 | 12.29785 |
| | 2 | 0.818 [g] | 0.669 | 0.631 | 10.47757 |
| | 3 | 0.890 [h] | 0.791 | 0.752 | 8.57908 |
| Latin America and the Caribbean | 1 | 0.859 [i] | 0.737 | 0.705 | 11.41227 |
| | 2 | 0.932 [j] | 0.868 | 0.830 | 8.65307 |
| | 3 | 0.970 [k] | 0.941 | 0.911 | 6.25703 |
| | 4 | 0.988 [l] | 0.977 | 0.959 | 4.26026 |
| | 5 | 0.999 [m] | 0.998 | 0.996 | 1.38816 |
| North America | 1 | 1.000 [n] | 1.000 | | |

[a]. Predictors: (Constant), Established Business Ownership; [b]. Predictors: (Constant), Established Business Ownership, Governmental Programs; [c]. Predictors: (Constant), Established Business Ownership, Governmental Programs, Governmental Support and Policies; [e]. Predictors: (Constant), Perceived Capabilities; [f]. Predictors: (Constant), Business Services Sector; [g]. Predictors: (Constant), Business Services Sector, Governmental Programs; [h]. Predictors: (Constant), Business Services Sector, Governmental Programs, Total Early-Stage Entrepreneurial Activity (TEA); [i]. Predictors: (Constant), Cultural and Social Norms; [j]. Predictors: (Constant), Cultural and Social Norms, Taxes and Bureaucracy; [k]. Predictors: (Constant), Cultural and Social Norms, Taxes and Bureaucracy, Commercial and Professional Infrastructure; [l]. Predictors: (Constant), Cultural and Social Norms, Taxes and Bureaucracy, Commercial and Professional Infrastructure, Business Services Sector; [m]. Predictors: (Constant), Cultural and Social Norms, Taxes and Bureaucracy, Commercial and Professional Infrastructure, Business Services Sector, High Job Creation Expectation; [n]. Predictors: (Constant), Governmental Programs.

For the final model with the dependent variable innovation capability, the influence statistics cannot be calculated because the fit is perfect for Africa and North America.

In each continent, we have to consider the R square of the superior model obtained in the stepwise method. Thus, the R Square for the best model obtained for each of the remaining five continents (Africa, Asia, Europe, Latin America and the Caribbean, and North America) is, respectively, 1.0, 0.515, 0.791, 0.998, 1.0, which represents satisfactory values. Then, we present, for each of the continents, the coefficients that allow writing the linear regression equation for the best model obtained (Table 5).

Based on the results in Table 5, we elaborate the linear regression equations for the dependent variable innovation capability for the five continents.

The linear regression equation for Africa is:

$$\text{Innovation capability} = -31.831 - 1.548 \times \text{Established Business Ownership} + 38.563 \times \text{Governmental Programs} + 0.638 \times \text{Governmental Support and Policies}$$

The linear regression equation for Asia is:

$$\text{Innovation capability} = 89.896 - 0.763 \times \text{Perceived Capabilities}$$

**Table 5.** Innovation capability coefficients.

| | | Coefficients | | | | | |
|---|---|---|---|---|---|---|---|
| **Continent** | | **Model** | **Unstandardized Coefficients** | | **Standardized Coefficients** | **t** | **Sig.** |
| | | | **B** | **Std. Error** | **B** | | |
| Africa | 1 | (Constant) | 48.352 | 1.632 | | 29.634 | 0.001 |
| | | Established Business Ownership | −1.623 | 0.103 | −0.996 | −15.818 | 0.004 |
| | 2 | (Constant) | −26.830 | 0.334 | | −80.429 | 0.008 |
| | | Established Business Ownership | −1.557 | 0.001 | −0.956 | −2202.426 | 0.000 |
| | | Governmental Programs | 36.875 | 0.164 | 0.098 | 225.481 | 0.003 |
| | 3 | (Constant) | −31.831 | 0.000 | | | |
| | | Established Business Ownership | −1.548 | 0.000 | −0.950 | | |
| | | Governmental Programs | 38.563 | 0.000 | 0.102 | | |
| | | Governmental Support and Policies | 0.638 | 0.000 | 0.005 | | |
| Asia | 1 | (Constant) | 89.896 | 10.056 | | 8.940 | 0.000 |
| | | Perceived Capabilities | −0.763 | 0.205 | −0.718 | −3.716 | 0.003 |
| Europe | 1 | (Constant) | 17.919 | 9.885 | | 1.813 | 0.087 |
| | | Business Services Sector | 1.654 | 0.376 | 0.720 | 4.397 | 0.000 |
| | 2 | (Constant) | −5.174 | 11.804 | | −0.438 | 0.667 |
| | | Governmental Programs | 13.776 | 4.933 | 0.464 | 2.792 | 0.013 |
| | | Business Services Sector | 1.075 | 0.382 | 0.467 | 2.813 | 0.012 |
| | 3 | (Constant) | −1.102 | 9.756 | | −0.113 | 0.911 |
| | | Governmental Programs | 17.069 | 4.180 | 0.575 | 4.083 | 0.001 |
| | | Business Services Sector | 1.086 | 0.313 | 0.472 | 3.471 | 0.003 |
| | | Total Early-Stage Entrepreneurial Activity (TEA) | −1.656 | 0.541 | −0.367 | −3.059 | 0.007 |
| Latin America and the Caribbean | 1 | (Constant) | −106.505 | 30.669 | | −3.473 | 0.008 |
| | | Cultural and Social Norms | 46.108 | 9.727 | 0.859 | 4.740 | 0.001 |
| | 2 | (Constant) | −125.956 | 24.402 | | −5.162 | 0.001 |
| | | Cultural and Social Norms | 39.868 | 7.748 | 0.743 | 5.146 | 0.001 |
| | | Taxes and Bureaucracy | 17.877 | 6.798 | 0.379 | 2.630 | 0.034 |
| | 3 | (Constant) | −227.399 | 41.283 | | −5.508 | 0.002 |
| | | Cultural and Social Norms | 34.700 | 5.916 | 0.646 | 5.865 | 0.001 |
| | | Taxes and Bureaucracy | 22.977 | 5.262 | 0.488 | 4.367 | 0.005 |
| | | Commercial and Professional Infrastructure | 37.686 | 13.865 | 0.296 | 2.718 | 0.035 |
| | 4 | (Constant) | −209.666 | 28.805 | | −7.279 | 0.001 |
| | | Business Services Sector | 0.547 | 0.194 | 0.244 | 2.818 | 0.037 |
| | | Cultural and Social Norms | 26.416 | 4.987 | 0.492 | 5.298 | 0.003 |
| | | Taxes and Bureaucracy | 23.268 | 3.584 | 0.494 | 6.492 | 0.001 |
| | | Commercial and Professional Infrastructure | 37.892 | 9.441 | 0.297 | 4.014 | 0.010 |
| | 5 | (Constant) | −188.064 | 9.946 | | −18.909 | 0.000 |
| | | Business Services Sector | 0.711 | 0.068 | 0.317 | 10.455 | 0.000 |
| | | Cultural and Social Norms | 32.113 | 1.842 | 0.598 | 17.433 | 0.000 |
| | | Taxes and Bureaucracy | 22.521 | 1.173 | 0.478 | 19.194 | 0.000 |
| | | Commercial and Professional Infrastructure | 27.017 | 3.494 | 0.212 | 7.733 | 0.002 |
| | | High Job Creation Expectation | −0.420 | 0.064 | −0.217 | −6.565 | 0.003 |
| North America | 1 | (Constant) | −2009.400 | 0.000 | | | |
| | | Governmental Programs | 643.333 | 0.000 | 1.000 | | |

The linear regression equation for Europe is:

$$\text{Innovation capability} = -1.102 + 17.069 \times \text{Governmental Programs} + 1.086 \times \text{Business Services Sector} - 1.656 \times \text{Total Early-Stage Entrepreneurial Activity (TEA)}$$

The linear regression equation for Latin America and the Caribbean is:

$$\text{Innovation capability} = -188.064 + 0.711 \times \text{Business Services Sector} + 32.113 \times \text{Cultural and Social Norms} + 22.521 \times \text{Taxes and Bureaucracy} + 27.017 \times \text{Commercial and Professional Infrastructure} - 0.420 \times \text{High Job Creation Expectation}$$

The linear regression equation for North America is:

$$\text{Innovation capability} = -2009.4 + 643.333 \times \text{Governmental Programs}$$

The application of linear regression verified the assumptions of normality demonstrated by the graphs of normal probability of the residuals and the Kolmogorov–Smirnov test, where *p*-values of 0.941, 0.308, 0.867, 0.06 were obtained for each of the continents (Africa, Asia, Europe, Latin America and the Caribbean).

Table 6 presents a summary of the results for innovation capability and business dynamism.

**Table 6.** Summary of results.

| Continents/ Variable | Linear Regression Equation |
|---|---|
| **Business Dynamism** | |
| Africa | Business dynamism = 69.755 − 2.299 × Established Business Ownership |
| Asia | Business dynamism = 25.794 + 12.197 × Financing for Entrepreneurs + 0.436 × Business Services Sector |
| Europe | Business dynamism = 0.845 + 0.694 × Business Services Sector + 0.520 × High Status to Successful Entrepreneurs + 24.248 × Female/Male TEA |
| Latin America and the Caribbean | Business dynamism = −188.913 + 31.101 × Cultural and Social Norms + 35.064 × Taxes and Bureaucracy + 22.388 × Post School Entrepreneurial Education and Training |
| North America | Business dynamism = −190.845 + 89.545 × Taxes and Bureaucracy |
| **Innovation Capability** | |
| Africa | Innovation capability = −31.831 − 1.548 × Established Business Ownership + 38.563 × Governmental Programs + 0.638 × Governmental Support and Policies |
| Asia | Innovation capability = 89.896 − 0.763 × Perceived Capabilities |
| Europe | Innovation capability = −1.102 + 17.069 × Governmental Programs + 1.086 × Business Services Sector − 1.656 × Total Early-Stage Entrepreneurial Activity (TEA) |
| Latin America and the Caribbean | Innovation capability = −188.064 + 0.711 × Business Services Sector + 32.113 × Cultural and Social Norms + 22.521 × Taxes and Bureaucracy + 27.017 × Commercial and Professional Infra-structure − 0.420 × High Job Creation Expectation |
| North America | Innovation capability = −2009.4 + 643.333 × Governmental Programs |

As seen in Table 6, and regarding business dynamism, Europe and Asia are the only continents that do not have variables that negatively affect business dynamism. However, Europe has more variables that positively affect business dynamism than Asia. Regarding Africa, Latin America and the Caribbean, and North America, it seems that Latin America and the Caribbean have more variables that positively affect business dynamism. The continent where business dynamism is most negative is North America.

Regarding innovation capability, we found that it is negative on all continents except in Asia. North America is the continent with the most negative innovation capability. Latin America and the Caribbean have more variables that positively affect innovation capability.

In the next section, we will discuss the results of the present research.

## 5. Discussion of Results

Following the obtained results, we will first discuss the linear regression equations with the dependent variable "business dynamism" and later with the variable "innovation capability" for the African, Asian, European, Latin America and the Caribbean, and North America continents.

### 5.1. Discussion of the Results of the Linear Regression Equation of the Dependent Variable "Business Dynamism"

Regarding the African continent, our results indicate that the variable "Business dynamism" is positive, however, the other variable that the model includes "established business ownership" negatively affects the model. According to [34] the competitive environment in emerging (African) markets cannot be characterized as a free market environment, however, a set of political measures has been implemented in order to improve this situation. Thus, the variables business dynamism, technological change, low-cost strategies, and accessibility marketing are the main drivers of the performance and competitiveness of these markets [34]. The company's "established business ownership" is an important step in the development of business and the economy. This is because it provides a more stable economy and creates more jobs. The managing partners of established companies contribute a lot to the society in which they operate, even if they are micro or small companies [35]. In this way, our results are in line with that indicated by [34,35].

Regarding the Asian continent, the variables "business dynamism", "financing for entrepreneurs", and "business services sector" positively affect the model. East Asia, with its many highly successful economies, has strong economic ties and business dynamism [36]. Regarding the variable "financing for entrepreneurs", as a rule, Asian countries face a great complexity of formalities on the part of financial institutions to obtain credit to start or continue with their companies in operation [37]. Our results point to the opposite of that indicated by [37]. The business services sector in Asian countries has grown in urban areas in the business services sector. Sales of food, sports, and entertainment have also been increasing, but not as much as the business services sector [38]. In this alignment, our results confirm that indicated by [38].

Regarding Europe, we found that the variable "business dynamism", "business services sector", "high status to successful entrepreneurs", and the female/male TEA positively affect the model. The variable that has a value that differs from the others is the "Female/Male TEA". The "business dynamism" variable has been declining throughout the present century, both in companies that have been in the market for a long time and in new companies. This regressive trend cannot be explained only with the most recent economic crisis. Globalization and the opening of new markets, make it possible for more efficient foreign companies to "steal" the market from domestic companies. The fear that foreign multinationals will monopolize domestic markets is one of the most important factors for the decrease in business dynamism [39]. According to [40] the effect of the entry of foreign companies can have a negative effect in the short term, but the same is reversed in the long term.

Domestic companies are able to learn and create partnerships with foreign companies. Foreign investment plays a very important role in business dynamism. Although business dynamism has been declining in Europe over the past few decades [41], it is still positive as confirmed by the present research. With regard to the variable "business services sector", during the last decades, there has been a rapid expansion, which has had positive impacts in Europe [42].

Our results reinforce the findings of [42]. Regarding the variable "high status to successful entrepreneurs", it is higher in the "leading" countries of Southern and Eastern Europe [43]. As can be seen in our research, high status to successful entrepreneurs is the variable that influences less business dynamism in Europe, taking into account the variables that enter the model. The last variable that appears in the model found for

Europe is the "Female/Male TEA". In Europe, with regard to gender issues, men and women are significantly associated with informal factors such as cultural and social norms, perceived opportunities to create an entrepreneurial social image, and formal factors such as intellectual property rights. Informal factors have a greater impact on entrepreneurial activity than formal factors [44], which is in line with our results because the value of the variable "Female/Male TEA" is the highest of the variables, that is, it is the variable that most influences business dynamism.

Regarding the discussion of linear regression for Latin America and the Caribbean, we found that the variable "business dynamism" negatively affects the model found. The remaining variables positively affect the model ("cultural and social norms"; "taxes and bureaucracy"; "post school entrepreneurial education and training"). The variable "business dynamism" comes with a very negative value, as there are many micro-companies that are informal. These microenterprises are seen as sources of business opportunities for some and sources of economic survival for others. However, informal micro-enterprises, as a rule, are positioned in a highly competitive market with many rivals and where price, product variety, and consumer credit generally differentiate successful micro-companies [45]. Our real results for business dynamism are explained by the fact that in Latin America and the Caribbean there are many informal micro-companies that are not counted in GEM. The variable "cultural and social norms" is important in the development of entrepreneurship, and culturally, in Latin America and the Caribbean, being an entrepreneur is recognized and held in high esteem which is consistent with the results of [15,46].

Regarding the "Taxes and Bureaucracy" variable, they are important indicators for the beginning of commercial activity. It is noticed that, at this moment in the countries of Latin America and the Caribbean, there are fewer taxes and less bureaucracy to start a business, comparing, for example, with countries in Europe [15]. Taxes and bureaucracy also positively influence business dynamism, as we can see in our results. The "post school entrepreneurial education and training" variable is important in the development of entrepreneurship, whether in the creation, management, or growth of businesses [47]. Our results are in line with those indicated by [47], while the authors of [15] indicate the opposite. With regard to the linear regression equation for North America, it appears that the variable "business dynamism" negatively affects the model found and "taxes and bureaucracy" positively affect the model. Thus, business dynamism has been decreasing, as the authors of [41,48] claim. North American countries with higher economic freedom have better rates of gross and net job creation as well as business creation. In this perspective, it is possible to suggest a relationship between freedom and business dynamism. This fact supports the theories by which government policies may inhibit business dynamism [49]. Our results confirm those indicated by [48–50]. With regard to the "taxes and bureaucracy" variable, our results indicate that these are reduced in the countries of North America. Taxes and bureaucracy are the biggest obstacles to entrepreneurship [51].

*5.2. Discussion of the Results of the Linear Regression Equation of the Dependent Variable "Innovation Capability"*

With regards to the linear regression equation for Africa, we find that the "innovation capability" negatively affects the model found. The remaining three variables (established business ownership; governmental programs; governmental support and policies) positively affect the model found. It should be noted that the variable "Governmental Support and Policies" is the one that influences the "Innovation capability" in a more relevant way. In Africa, innovation capability has an uneven distribution in its different countries [52]. According to the authors of [53], the combination of three factors (technological innovation capacity; available basis (technological infrastructure and human capital); and protection and intellectual property patents), plays an important role in the positioning of each national innovation system. Thus, our results are complementary to those indicated by [53,54]. The "established business ownership" variable is not assumed to be one of the key predictors for economic growth in African countries such as South Africa [55].

The results of [55] are in line with our results because established business ownership negatively influences innovation capability. The "governmental programs" variable assumes great importance in the "innovation capability". As a rule, African countries, despite having natural resources of great economic value, many of them are still extremely poor. Governments have tried to increase the complexity of governmental programs by making them more transparent processes [56].

However, some countries, such as Botswana, have successfully implemented relatively free-market policies that we can associate with the variable "governmental support and policies". With low taxes and disciplined government spending, Botswana's growth was much faster [56]. Our results reinforce what was indicated by [56].

Moving on to the discussion of the results of the linear regression equation for Asia, it appears that the variable "innovation capability" positively affects the model, in contrast, the variable "perceived capabilities" negatively affects the model. The development of innovation capability in companies and industries in emerging Asian economies has faced significant challenges due to a lack of knowledge [57]. This conclusion is opposite to our results because the innovation capability has very positive values. Our results are in line with [58]. The authors of [58] state that industrial actors are able to obtain resources and incentives to facilitate their production and investments in innovation, thus improving local innovation institutions. There are factors that affect the perceived capabilities. Factors such as perceived opportunities in the country, the fear of failure, intellectual property rights, and entrepreneurship as a career option, knowledge transfer rate, and overall performance in the higher education system have a substantial impact on the perceived capabilities [59]. Our results indicate that the factors identified by [59] negatively influence perceived capabilities and consequently innovation capability.

Regarding the linear regression equation for Europe, we found that the variables "innovation capability" and "TEA" negatively affect the model found. On the contrary, there is the variable "governmental programs" and the "business services sector" variable. According to the authors of [60], in Europe (Spain), knowledge sharing is essential to improve the companies' innovation capability. However, depending on the size of the company's innovation capacity and technological intensity, the type of knowledge sharing that appears to be most productive varies. The intensity of the technology also moderates the degree of relevance of each innovation capability in creating value. Our results are complementary to that indicated by [60]. The "governmental programs" variable has a positive impact on innovation capability. In 2014, Europe changed its policies based on the smart specialization domains of each region. These new policies are called RIS3 (research and innovation strategies for smart specialization [61,62]. Our results suggest that RIS3 has a positive impact on innovation capability in Europe. The variable "business services sector" has a positive impact on the model found, which is in line with that indicated by [63]. The European economy, in general, specializes in the "business services sector". In France, the weight of the business services sector in GDP has been increasing [64]. The "TEA" variable negatively influences innovation capability, which is in line with what was indicated by [65]. Innovative entrepreneurs are more present in countries with higher levels of development and income, being motivated by the opportunity they see in becoming entrepreneurs [66].

Regarding the linear regression equation for Latin America and the Caribbean, the results indicate that the variable "innovation capability" and "high job creation expectation" negatively affect the model found. The remaining variables (business services sector; cultural and social norms; taxes and bureaucracy; commercial and professional infrastructure) positively affect the model found. Our results regarding the "innovation capability" demonstrate that it has a very negative value. This value can be justified by the results of [67]. The author found that bribery had a significantly negative impact on the innovation capacity and productivity of the companies observed. The author of [67] defined bribery as informal payments made by companies to public officials to "get things done". The innovation strategy, the hiring and professional development policies, and the external

structure are the most relevant factors for the creation of new ideas and for the management of innovation projects [68]. The "business services sector" variable has achieved gains in market coverage. The service sector is increasingly sophisticated, which is evidenced by the growing strength of the international trading system [69]. Our results confirm this trend as the variable business services sector positively affects innovation capability. With regard to "cultural and social norms", it appears that it has a positive effect on innovation capability. As a rule, traditions, religion, the lifestyle and the "friendliness" of the inhabitants of Latin America make them more likely to share with others, and the ability to achieve harmony in social relationships [70], which explains our results. "Taxes and bureaucracy" positively influence innovation capability and governments must therefore maintain the policies in force. Our results are complementary to that indicated by [71].

The authors state that taxes and bureaucracy are appropriate for the process of promoting and improving business activities. The "commercial and professional infrastructure" variable positively affects innovation capability.

However, the authors of [15] indicate that Latin American countries need to improve commercial and professional infrastructure, as it is still low compared to Western European countries. The "high job creation expectation" variable, according to our results, negatively influences innovation capability. No studies were found that relate these two variables. However, some authors indicate that there is a negative relationship between high job creation expectations and economic competitiveness [72,73].

Moving on to the discussion of the results of the linear regression equation for North America, it was found that the variable "innovation capability" has a very negative value in the model found. The opposite is the variable "governmental programs". In this respect, three variables may considerably affect the quality of the country's innovation capability: (1) the flexibility given to companies located in the country to recruit and assign employees to the most productive assignments; (2) business networks and support industries in the country; (3) the quality of public institutions in the country where the main economic actors interact [74]. As the innovation capability variable has a very negative value, we can say that according to the results of [74], the three variables that influence innovation capability must have negative values. The "governmental programs" variable positively affects innovation capability, which complements [75] results. The author indicates that the national R&D system and governmental programs contribute strongly to the development of the industry. Some government programs have tried to encourage entrepreneurial activity by making loans available to new startups that they believe will be successful [76].

*5.3. Business Dynamism and Innovation Capability—Interactions with Open Innovation*

In a context where it is intended to study the influencing aspects of entrepreneurship on a global scale, the analysis of indicators such as business dynamism and innovation capability is absolutely central, as was evident throughout the present research.

Nevertheless, the context of social, political, and economic complexity in which modern societies live makes it essential that we relate these variables with open innovation. This term was popularized in 2003 by Henry Chesbrough, who conceived the idea of collaboration between people and entities outside the organization.

The central idea of open innovation is that valuable ideas may come from the inside or outside of the company and reach the market [77]. In reality, the idea of openness implies breaking with the traditional philosophy of not revealing internal knowledge to the outside of the organization. This attitude of openness to the outside implies recognition to obtain competitive advantages, either through relationships with other professionals or through interaction with other external organizations. From this relationship come resources and knowledge share, as well as the establishment of alliances and partnerships, thus obtaining an acceleration in the innovation processes at the various levels of the management of organizations.

However, although the focus of open innovation may be on the side of companies or organizations, business dynamism and innovation capability can clearly be promoted

through the implementation of policies that are against an environment favorable to the increase of successful entrepreneurship [78]. From this research, public sectors and governments, all over the world, have at their disposal several resources that they can use in order to create a culture of open innovation, which promotes business dynamism and innovation capability. When combining the results of business dynamism and innovation capability, Asia is the continent that stands out, which suggests that open innovation is more developed and implemented in the corporate culture of companies in that continent. Moreover, the results suggest that the closest continent to Asia is Europe. Nevertheless, Europe has to increase TEA and innovation capability. Africa has business dynamism well implemented, but they are unable to have the capacity for innovation, which undermines open innovation. Africa's companies and policymakers need to focus on improving established business ownership and innovation capacity. Latin America and the Caribbean and North America have negative results in business dynamism and innovation capability. Policymakers, companies, and academia have to work together and develop measures that focus on improving these two indicators. The implementation of open innovation policies can help to speed up the process. This situation may be due to the fact that there are many relaxed businesses that contribute to the growth of the informal economy [15]. Policymakers have to implement measures to reduce the informal economy. With these measures, open innovation will certainly improve. In this context, political decision-makers have the ability to create laws, projects, network promotion, voluntary cooperation, public-private partnerships, public consultation for problem-solving, the approach of universities to the business network, creation of markets, common distribution and commercialization channels, creation of common physical or virtual spaces, incubators or industrial zones, promotion of strategic alliances, sharing of patents and technologies, amongst others [78].

The results suggest that the promotion of these or other open innovation initiatives will be as decisive for the promotion of entrepreneurship as the levels of business dynamics and innovation capacity that companies, countries, or even the various continents here studied may represent.

To understand which types of initiatives, strategies, or policies make more sense to implement in each continent, we present below the conclusions of the research, with a focus on its practical contributions.

## 6. Concluding Remarks

This research aims to present the main factors that best explain business dynamism and innovation capability for the continent of Africa, Asia, Europe, Latin America and the Caribbean, and North America. Being so, the target data for analysis were collected in the GEM and GCR of 2018. As a result, linear regression models for the dependent variables business dynamism and innovation capability of the five continents mentioned above were arrived at.

### 6.1. Practical Implications

The models detected in the African continent have positive business dynamism and negative innovation capability. In order to reverse the negative trend of innovation capability, business leaders should consult innovation studies to better understand technological advances. Regarding open innovation, entrepreneurs should access information on technological capabilities in different countries, in order to understand the geographical context in which companies can develop and establish their innovation activities [52,79,80]. Established business ownership negatively influences business dynamism and innovation capability, thus being essential to create ecosystems that encourage innovation, where universities play a crucial role.

Regarding the Asian continent, it appears that in the models detected, business dynamism and innovation capability are positive. From the continents under investigation, Asia is where open innovation is best implemented. However, the perceived capabilities may negatively influence innovation capability, this can be explained by the fact that in

some Asian countries, there are increasing cases of drug and human trafficking, organized crime, and illegal migration. In this context, more measures should be implemented related to security, operations to combat terrorism, education, water resources management, and energy efficiency, thus contributing to the creation of new values in civil society [81].

Regarding the models found in the European continent, it was discovered that business dynamism is positive and innovation capability negative. In order for innovation capability to be positive, allowing to improve dynamics and enhance open innovation, the focus of the policy should be on the development of rules and regulations to expand the quality of public institutions, the quality of business networks and support industries, and flexibility in the labor markets should be emphasized in its countries [74]. Innovation capability is also negatively influenced by TEA. Europe needs the culture of entrepreneurship and free enterprise to become dominant and to fight resistance to change and protect established interests. In Europe, companies have to form more knowledge networks so that the innovation capability may naturally grow. Our results indicate that RIS3 has a positive impact on innovation capability in Europe.

Regarding the models found for Latin America and the Caribbean, it appears that business dynamism and innovation capability are negative, which will clearly affect open innovation negatively. According to the authors of [82], the reason for succeeding, as indicated above, is that smart growth has shown itself to be very concentrated and its urban development policies are less and less flexible. Policymakers assumed that these policies would meet social and environmental needs but did not make them explicit. It was also found that the variable High Job Creation Expectation negatively influences innovation capability. In order for business dynamism and innovation capability to be positive, the results suggest that the domains of specialization of smart growth are redefined. Regional development policies must be made more flexible to meet social and environmental needs. For the innovation capability to show positive numbers, governments must implement measures to control bribes. Increasing innovation capability can help to control corruption [67,83]. It is also suggested to implement stronger disciplinary audits, strictly impose the legal supervision of government officials, and increase penalties for violating disciplinary laws [84].

Finally, the models found for North America show that business dynamism and innovation capability are negative, which can negatively affect open innovation. In order for these variables to reach positive numbers, governments can implement new policies (despite the fact that Governmental Programs positively influence innovation capability) such as giving tax incentives to companies for the first purchase of important equipment and high technology products with intellectual property rights developed by domestic companies and provide support policies directed to companies that purchase domestic high-tech equipment [85].

This research contributes measures for organizations and policymakers to apply to five continents (Africa, Asia, Europe, Latin America and the Caribbean, and North America), with the purpose to improve the performance of business dynamism and innovation capability.

### 6.2. Limitations and Future Research Lines

The present research has limitations inherent to the quantitative methodology: it does not always clarify the entire complexity of human experience or perceptions; can reveal what/to what extent but cannot always explore why or how; may give a false impression of sample homogeneity. Thus, the results of this research should be further developed with qualitative studies. The data used for analysis were collected in GEM and GCR on 15 November 2018, so data added later are not included in this research.

Regarding future lines of research, other methodological approaches should be considered to demonstrate the relationships between variables that can explain business dynamism and innovation capability in different contexts, such as structural equations or path analysis. Complementary studies should be carried out to verify whether RIS3 has a positive impact on innovation capability in Europe. The business dynamism and

innovation capability present very negative values in the models found in Latin America and the Caribbean and North America and it is necessary for future studies to find out why.

**Author Contributions:** Conceptualization, J.L., M.O., P.S., L.F. and J.O.; methodology, J.L., P.S. and L.F.; software, P.S. and L.F.; validation, M.O. and J.O.; formal analysis, J.L., P.S. and L.F.; investigation, J.L., M.O. and J.O.; resources, J.O.; data curation, P.S. and L.F.; writing—original draft preparation, J.L., M.O. and J.O.; writing—review and editing, J.L. and M.O.; visualization, M.O.; supervision, J.L.; project administration, J.L. All authors have read and agreed to the published version of the manuscript.

**Funding:** This research received no external funding.

**Conflicts of Interest:** The authors declare no conflict of interest.

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
