# Peer review of "Business Dynamism and Innovation Capacity, an Entrepreneurship Worldwide Perspective"

_2199-8531, doi:10.3390/joitmc7010094_

Round 1

Reviewer 1 Report

Authors must make the following corrections in the paper:

-  Authors should explain better the academic contribution of the work developed. Highlighting what is innovative / original about the existing literature.

- Authors should develop the Abstract, presenting in particular the main contributions / results of the work.

- Figures 1 and 2 are not very clear. Mainly figure 1.

- The authors must explain how the countries indicated in table 3 were chosen.

-The authors must explain the reason for choosing the linear regression method in this work

Author Response

Dear reviewer,

Reviewer 2 Report

Dear authors, thank you very much for your submission to the Journal of Open Innovation and for giving me the opportunity to review your interesting manuscript. Your research is appreciated as it sheds light on an interesting research gap. However, I found several issues that need to be addressed before publication in JOI can be considered:

  1. Entrepreneurship should be in the title.
  2. You abstract needs to be improved. Please clearly state the research question and your findings. Your results should all be presented in the same manner and be clear answers to your research question.
  3. The same applies to the introduction. Presenting explanatory models for certain regions with regard to business dynamism and innovation capacity is no goal. What do you want to explain?
  4. 2: Better use Literature Review than Revision.
  5. Rather than explaining entrepreneuership and presenting a long Tables 1 and 2, you need to explain business dynamics and innovation capacity.
  6. You should visualize your explanatory model(s). What are the (in)dependent variables, moderators, and/or mediators and how do they relate?
  7. You should present your final results in a table: What are the differences between the countries?

I hope you find my comments helpful. Good luck with your revision!

Author Response

Dear reviewer,

Round 2

Reviewer 2 Report

I appreciate your efforts you invested in your revision.